# Impact of HCV Infection on Hepatocyte Polarity and Plasticity

**DOI:** 10.3390/pathogens11030337

**Published:** 2022-03-10

**Authors:** Jean Agnetti, Christophe Desterke, Ama Gassama-Diagne

**Affiliations:** 1INSERM, UMR-S 1193, Université Paris-Sud, F-94800 Villejuif, France; jean.agnetti@inserm.fr; 2UFR Médecine-INSERM UMS33, Université Paris-Sud, F-94800 Villejuif, France; christophe.desterke@inserm.fr

**Keywords:** HCV, hepatocytes, polarity, EMT, cell junctions, traffic, ECM

## Abstract

The hepatitis C virus (HCV) is an oncogenic virus that alters the cell polarization machinery in order to enter the hepatocyte and replicate. While these alterations are relatively well defined, their consequences in the evolution of the disease remain poorly documented. Since 2012, HCV infection can be effectively cured with the advent of direct acting antivirals (DAA). Nevertheless, patients cured of their HCV infection still have a high risk of developing hepatocellular carcinoma (HCC). Importantly, it has been shown that some of the deregulations induced by HCV are maintained despite a sustained virologic response (SVR), including the down-regulation of some hepatocyte functions such as bile acid metabolism, exemplifying cell dedifferentiation, and the up-regulation of the epithelial–mesenchymal transition (EMT). EMT is a process by which epithelial cells lose their differentiation and their specific polarity to acquire mesenchymal cell properties, including migration and extracellular matrix remodeling capabilities. Of note, epithelial cell polarity acts as a gatekeeper against EMT. Thus, it remains important to elucidate the mechanisms by which HCV alters polarity and promotes EMT that could participate in viral-induced hepatic carcinogenesis. In this review, we define the main steps involved in the polarization process of epithelial cells and recall the essential cellular actors involved. We also highlight the particularities of hepatocyte polarity, responsible for their unique morphology. We then focus on the alterations by HCV of epithelial cell polarity and the consequences of the transformation of hepatocytes involved in the carcinogenesis process.

## 1. Introduction

The hepatitis C virus was first identified in 1975 as a causative agent of non-A and non-B hepatitis [1] while its genome was cloned in 1989 [2]. This virus infects and replicates almost exclusively in hepatocytes and is a major causative factor in many liver diseases such as steatosis, fibrosis, cirrhosis and hepatocellular carcinoma [3,4,5,6]. There is still no vaccine available against HCV, and according to the World Health Organization (WHO), 58 million people are still infected with HCV, and there are 1.5 million new infections each year. Furthermore, WHO estimated that in 2019 nearly 290,000 people died from diseases caused by chronic HCV infection, mainly from cirrhosis and cancer. Furthermore, whereas HCV is a hepatotropic virus that primarily affects the liver, several reports have linked HCV to a variety of extrahepatic symptoms involving the skin, as well as the musculoskeletal, renal, cardiovascular, and nervous systems [7]. Notably, it was reported that up to 74% of chronic HCV patients suffer from at least one extrahepatic manifestation [8].

In 2020, the Nobel Prize in Medicine was given to Harvey J. Alter, Michael Houghton and Charles M. Rice for their work in identifying the virus and the mechanisms by which it infects and replicates in hepatocytes. These works provided the basis for the development of direct-acting agents (DAA), the first of which obtained market authorization in 2011 with an efficacy of more than 90% in inducing a sustained virological response (SVR), compared to approximately 50% with the previous reference treatment, interferon α.

Nevertheless, while the infection can be cured, recent works have highlighted that the risk of HCC occurrence remains [9,10] and some HCV-induced deregulations are maintained despite the complete clearance of the virus [10,11,12]. Notably, Hamdane et al. showed that 2193 genes with epigenetic and transcriptional modifications were maintained despite eradication of the virus by DAAs [8]. Furthermore, in this study, 1411 of these genes were found to be closely related to the prognostic liver signature (PLS), a well-characterized liver expression signature of 186 genes that has been shown to predict survival and HCC risk in patients with advanced liver disease, regardless of etiology, including 73 poor prognosis/high HCC risk genes and 113 good prognosis/low HCC risk genes [13,14]. Thus, it is likely that the maintenance of some deregulation in patients treated with DAA may contribute to a worsening of the clinical picture regarding HCC. Among the processes associated with genes that are still deregulated, EMT remains up-regulated while hepatocyte specific functions such as xenobiotic and bile acid metabolism remain down-regulated, illustrating a dedifferentiation of hepatocytes [10]. EMT is a complex and reversible process by which epithelial cells lose their polarity, epithelial characteristics and differentiation to acquire a mesenchymal phenotype with invasive properties that promote metastasis [15,16]. Importantly, the polarization of epithelial cells represents a barrier against their dedifferentiation and EMT [17,18]. It is therefore essential to understand the mechanisms involved in hepatocyte polarization in order to determine how these processes are deregulated by HCV and could promote cancer even after HCV eradication from the infected liver. Importantly, epigenetic changes are not only due to a direct action of HCV on hepatocytes but also to an indirect action mediated by HCV’s modification of the environment surrounding the hepatocytes [10]. Thus, it is also important to define how HCV establishes a deleterious environment around hepatocytes, conducive to their transformation.

In this review, we will define the main actors involved in the polarization of epithelial cells and we will introduce the particularities of hepatocytes polarity responsible for their unique morphology. We will then focus on the mechanisms by which HCV alters hepatocyte polarity in order to enter the hepatocyte and replicate. Finally, we will describe the consequences of these alterations on hepatocyte plasticity, which could participate in liver carcinogenesis.

## 2. HCV Proteins

HCV is a positive strand RNA oncogenic virus of the Flaviviridae family. This virus is characterized by high genetic variability, with seven identified genotypes and 67 subtypes [19]. HCV mainly infects the liver and uses specific attachment and entry factors to cross the plasma membrane of hepatocytes and release its genomic RNA. Once released, the RNA serves as a template for replication and translation at the rough endoplasmic reticulum (ER) by the viral and host translation machinery. The 9.6-kilobase genome encodes a large polyprotein that is processed into three structural proteins, the core capsid protein and the envelope proteins E1 and E2, and at least seven non-structural proteins (NS proteins), NS1 (also called p7), NS2, NS3, NS4A, NS4B, NS5A, NS5B. The core is a 191 amino acid protein that makes up the nucleocapsid. It has been shown to bind lipid droplets (LD), the fatty acid storage organelles, and this interaction is required for viral replication [20,21]. Furthermore, core protein alters many cellular processes including transcription, lipid metabolism and apoptosis. The E1 and E2 proteins form the envelope and are involved in the entry of the virus into the hepatocytes. Non-structural proteins are required for virus replication. For example, NS5A expression induces the formation of double membrane vesicles (DMV) [22]. DMV are considered as the organelle where HCV replication takes place, in association with LD [23,24]. Note that although expression of NS5A alone is able to induce the formation of these DMV, expression of the full HCV genome significantly increases the amount of DMV formed [22]. In addition, NS5A modulates the response to interferon, thus conferring resistance to this molecule. This was considered as the gold standard treatment until the advent of DAA, which targets nonstructural proteins of the virus.

HCV proteins work together to set up a deleterious environment for hepatocytes that can lead over a long period of time, from 20 to 40 years, to the establishment of hepatocellular carcinoma (HCC). Notably, HCV infection alters gene expression, metabolism, signaling pathways, polarity of the hepatocytes and the extracellular environment surrounding these cells.

## 3. Hepatocyte Polarity

The liver contains five main cell types. The endothelial cells lining the blood vessels, the Kupffer cells, resident macrophages of the liver, the stellate cells regulating the extracellular matrix and two types of epithelial cells: the cholangiocytes lining the bile ducts and the hepatocytes. Hepatocytes are polarized epithelial cells representing nearly 70% of the total liver cells and constituting the liver parenchyma. These cells are involved in most of the metabolic and synthesis functions of the liver and also transport a wide range of endogenous or exogenous substances from the blood to the bile canaliculi. As for all epithelial cells, the establishment of hepatocyte polarity is essential for their terminal differentiation and their ability to perform their multiple functions. Epithelial cells exhibit an apico-basal polarity with an apical domain facing the lumen of the tube and the basolateral membranes that binds to the neighbor cells and the extracellular matrix (ECM) [25].

The establishment of apico-basal polarity is initiated by cascades of signaling pathways from cell–cell and cell–ECM junctions. These cues cause changes in the intracellular cytoskeleton, which organizes the cell cortex, and thereby the polarized endosomal trafficking leading to the formation of the apico-basal axis and lumen [26] The polarity complexes Crumbs (composed of the three proteins: Crumbs/proteins associated with Lin seven (PALS)/tight junction associate protein (PATJ)), Par (partition-defective 3 (Par-3)/partition-defective 3 (Par-6)/Atypical protein kinase C (aPKC)) and Scribble (Scribble/Discs large homolog 1 (Dlg1)/lethal giant larvae (Lgl)) are involved in the regulation of the cytoskeleton, cell trafficking, cell–cell junctions, required for epithelial cell polarization [25,27]. Of note, phosphoinositides and enzymes involved in their metabolism such as Phosphatase and TENsin homolog (PTEN), Phosphoinositide 3-kinase (PI3K) and SH2 domain containing inositol 5-phosphatase 2 (SHIP2) are also essential for polarization, notably for the establishment of apical and basolateral membrane identities, respectively, as depicted in Figure 1 [28,29,30].

As for all epithelial cells, the hepatocytes’ plasma membrane (PM) is divided into apical (canalicular) and basolateral (sinusoidal) domains but they are unique in that each hepatocyte is multi-polarized with several apical domains in mature liver [31]. Hepatocytes are in contact with the ECM at two of their opposite ends and multiple apical domains in one cell, unlike the columnar epithelial cells (Figure 1). This particularity contributes to their unique polarity, forming their lumens between two neighboring cells [32,33]. Furthermore, the existence of several apical poles in hepatocytes is partly due to a particular orientation of the mitotic spindle during cell division. Indeed, epithelial cells with columnar polarity divide symmetrically, sharing the apical and basolateral domains equally between the two daughter cells. In contrast, in hepatocytes, this cell division is described as asymmetric and allows the differential segregation of the apical domain in the daughter cells [34,35]. The organization of intracellular trafficking is also particular in the hepatocyte, with most of the canaliculi proteins indirectly addressed to the apical membrane by transcytosis (indeed the proteins are first addressed to the basolateral membrane and then targeted to the apical membrane), except for some bile acid transporters of the ABC and MRP family involved in the bile constituents secretion [36,37].

HCV uses the cell polarization machinery of hepatocytes to enter and replicate in these cells. While these alterations are relatively well defined, the consequences of these alterations in the evolution and the occurrence of steatosis, fibrosis, cirrhosis and hepatocellular carcinoma remain poorly documented.

## 4. HCV Infection and Cell–Cell Junctions

The characterization of HCV infection and replication mechanisms has been greatly facilitated by the development of in vitro cell culture models. Two-dimensional (2D) culture of Huh7.5 has long been considered a good model for studying HCV infection, due to its ease of use and permeability to HCV. While the permeability of Huh7.5 cells grown in 2D to the virus made this model attractive to researchers, it also constitutes its main limitation. Indeed, the entry of HCV in 2D Huh7.5 cells does not reflect the complexity of the mechanisms involved in vivo. This is mainly due to the low polarization of Huh7.5 cells during their culture in 2D. In 2D, Huh7.5 polarize little and the occludin and claudin proteins, necessary for HCV entry, are distributed uniformly along the plasma membrane. These proteins are therefore easily accessible by the virus, facilitating its entry into the cell compared to what happens in vivo in polarized hepatocyte. Notably, polarization of HepG2 cells has been shown to limit HCV access to the tight junction and thus HCV infection in these cells [38,39,40]. Therefore, the use of a 3D cell culture system that more accurately reflects the in vivo organization of hepatocytes has allowed us to learn more about the entry process of HCV. To enter into hepatocytes, HCV first interacts with CD81/tetraspanin-28 at the basolateral plasma membrane, via its envelope proteins, and then subsequently diffuses along the membrane to interact with the tight junction proteins’ claudins and occludins (Figure 1) [41,42,43]. Claudin-1 is essential for viral entry into the hepatocyte [40,44,45,46,47,48]. Indeed, depletion of this protein in the HCC-derived cell line Huh7 significantly decreases viral entry, whereas overexpression of certain claudins allows HCV entry into initially non-permissive lineages [45].

It was reported that HCV disturbs the adherens junctions. Particularly, HCV core protein induces downregulation of the gene encoding E-cadherin at transcriptional and post-translational level. Overexpression of HCV core protein in HepG2 cells induced the transcriptional downregulation of E-cadherin mediated by the activation of DNA methyltransferases 1 and 3b [49]. The HCV core protein also upregulates E12/E47 by inhibiting their ubiquitin-dependent proteasomal degradation [50]. These two proteins, which result from the alternative splicing of the E2A gene, repress E-cadherin expression by binding to its promoter [49,51]. HCV also alters polarity complex Scribble. Indeed, the NS4 protein of HCV associates with the Scribble protein to induce its degradation by the proteasome [52]. The core protein of HCV also disrupts the localization of the Scribble complex by inhibiting SHIP2, the enzyme responsible for the production of PtdIns(3,4)P2, which allows the anchoring of the Dlg1 protein to the plasma membrane [28,53]. These deregulations of the Scribble complex leads to disorganization of β-catenin at adherens junctions [28].

Importantly, depletion of E-cadherin leads to the destabilization of tight junctions in hepatocytes, notably, by disrupting the plasma membrane localization of occludins and claudins [54]. Thus, the destabilization of these cell–cell junctions following HCV infection limits the superinfection of hepatocytes, which could lead to their death [54,55,56]. This negative feedback would represent an interesting hypothesis explaining the capacity of this virus to establish itself in the long term during chronic infection. Of note, it was also reported that HCV also alters the integrity of tight junctions in a VEGF-dependent manner [57].

## 5. HCV Infection and Intracellular Trafficking

After entering the cell at the tight junctions via its clathrin-dependent endocytosis, HCV alters intracellular trafficking to facilitate its replication. The cytoskeleton and the small GTPases family of proteins play an essential role in intracellular trafficking [58]. The cytoskeleton is composed of three main types of filaments including microtubules responsible for most of the transport functions in the cells, thanks to the motor proteins associated with it (kinesin and dynein), actin microfilaments and intermediate filaments that are essential to the structure of the cell and the cohesion of the epithelial tissue [59]. To these three large families that compose the cytoskeleton, we can add the septins, GTP-binding proteins that are now considered as the fourth component of the cytoskeleton [60].

HCV alters intracellular trafficking to facilitate its replication and the traffic of LD is impacted in particular. LD are lipid storage organelles that consist of a core of neutral lipid surrounded by a monolayer of phospholipids and several proteins involved in their intracellular trafficking [61]. HCV infection leads to LD accumulation near the nucleus and also induces rearrangements of the intracellular membranes in a double membrane structure named the membranous web [62]. This creates a platform to facilitate HCV replication and the production of viral particles as well as their egress or secretion.

HCV alters the trafficking of the LD by modifying several components of the cytoskeleton. For example the core protein of the virus induced a microtubule- and dynein-dependent relocation of LD to the perinuclear region [63]. HCV also alters septins homeostasis, increasing the expression level of some genes encoding septins, particularly septin 9, which participates in the accumulation of LD at the perinuclear level [64]. Indeed, the depletion of septin 9 has been shown to decrease the number and the size of LD induced by HCV infection and also controls its replication [64].

The concentration of LD in the vicinity of the virus replication region is also related to the exploitation of the Ras-related protein Rab-18 by HCV viral proteins. Rab-18 is a protein belonging to the small GTPase Rab family (Ras superfamily). This family of proteins regulates many steps of cell traffic such as vesicle formation, trafficking and membrane fusion. Furthermore, Rab-18 is essential for LD homeostasis [65] and it was reported that core protein trafficking at the LD was dependent on Rab-18 [66]. In addition NS5A, localized at the double vesicle membranes (site of replication) [22] is able to bind Rab-18. Thus, Rab-18 is involved in bringing LD and DMV together, facilitating viral replication [67].

## 6. HCV Infection and ECM

ECM plays an essential role in the specific polarity and plasticity of hepatocytes and the formation of bile canaliculi. In vitro experiments quickly showed that the culture of primary hepatocytes on plastic resulted in the rapid loss of both their morphological and their functional properties [68]. In contrast, their functions could be maintained for a longer period of time while embedded in ECM such as matrigel or collagen I [69,70]. Furthermore, the hepatoblastoma-derived HepG2 cells grown in 2D on glass will form small apical structures shared between two to three cells. However, when these cells are cultured in three dimensions (3D) on ECM, they organize themselves in a multilayer structure that forms an elongated lumen—a structure closer to that found in vivo [71]. The ECM thus plays an important role in the polarity and differentiation of hepatocytes.

HCV induces hepatic fibrosis characterized by the remodeling of ECM surrounding the hepatocytes, which become denser and has a different composition than the matrix under physiological conditions [72]. These modifications have an effect on the polarity, bile canaliculi and differentiation of hepatocytes. Of note, the ECM surrounding hepatocyte is thinner and less dense than the basal lamina found in most other epithelia. In healthy mouse liver, the rigidity of the ECM adjacent to the hepatocytes is of the order of 150 Pa, measured by atomic force microscopy [73] whereas in a DDC-induced fibrosis model, the rigidity of the ECM can reach nearly 6000 Pa [73]. The increase in ECM rigidity has important effects on the differentiation status of hepatocytes. Indeed, culture of primary hepatocytes on a matrix of 1000 Pa or more leads to a decrease in albumin production and HNF4a gene expression level [73]. Mechanistically, the repression of the HNF4a gene is mediated by the small Rho GTPase and its effector, the Rho-associated protein kinase (ROCK), and to a lesser extent by focal adhesion kinase (FAK). It has been shown in vitro that primary human hepatocytes cultured on a 4600 Pa matrix express twice as much RhoA and less than half the albumin and HNF4a than the same cells cultured on a 600 Pa matrix after 7 days of culture [74]. Of note, pharmacological inhibition of ROCK using Y-27632 rescues HNF4a expression in a 1000Pa stiffness matrix [74]. Thus, matrix stiffness is a major regulator of hepatocyte plasticity.

## 7. Text-Mining Analysis of HCV Infection and Hepatocyte Polarity

In order to present a comprehensive review of the interactions between HCV and the hepatocyte cell polarization machinery, we performed a text-mining bioinformatics analysis (Figure 2).

Through this study, we searched for keywords associated with both HCV and cell polarization. Interesting processes and proteins emerged from this study, whose importance for virus entry, replication and alteration of the polarization machinery by the virus were detailed in this review. In particular, proteins such as CD81/tetraspanin and tight junction proteins occludin and claudin, required for virus entry into the hepatocyte were highlighted. Steatosis, fibrosis, cirrhosis and hepatocellular carcinoma also appeared among these keywords, illustrating a link between HCV-induced polarity alteration and the long-term consequences of HCV infection (Figure 2a).

We then analyzed the proteins associated with both HCV and the different components of polarity (Figure 2b). Proteins associated with HCV and polarity are involved in processes such as virus receptor activity, cell adhesion molecule binding and cadherin binding (Figure 2b,c). Proteins associated with the terms HCV and ECM are involved in tubulogenesis and tube development (Figure 2b,d). Interestingly, TGFβ, Zeb, metalloproteases are among the proteins associated with tubulogenesis. These proteins are also key players in the EMT process, suggesting that the alterations in cell polarization induced by HCV could be associated with EMT (Figure 2d). Finally, proteins associated with HCV and trafficking are involved in the viral process and the positive regulation of transport (Figure 2b,e).

## 8. HCV Infection and EMT

The alterations of essential actors involved in the establishment and maintenance of apico-basal polarization by HCV will participate in the progressive transformation of the hepatocyte, mainly by EMT, which could accompany the evolution of the disease from chronic HCV infection, steatosis to fibrosis and HCC.

Junctions and polarity complexes act as tumor suppressors [75]. Notably, these actors act as gatekeepers against EMT [17,18]. The transcription factor Snai1 is a potent inducer of EMT, inhibiting, among others, the transcription of the genes encoding E-cadherin, claudin 9 and cytokeratin 8 [76,77]. In polarized cells (and only in polarized cells), aPKC-mediated Snai1 phosphorylation promotes Snai1 protein degradation, thus preventing EMT [17]. As a consequence of cell junction and polarity complexes’ alteration induced by HCV, infected hepatocytes are more prone to EMT, which may have implications in hepatic carcinogenesis. Scribble delocalization from the plasma membrane to cytoplasm, as induced by HCV core protein [28], supports liver tumor formation and tumor cell invasiveness [78]. Notably, Scribble localization was shown to be disrupted in HCC, moving from the membrane to the cytoplasm. Furthermore, expression of a scribble mutant with a preferential cytoplasmic localization (ScribP305L) contributed to c-MYC-induced tumor development in murine liver.

HCV infection is a major cause of hepatic steatosis, a disorder characterized by an accumulation of LD in the hepatocytes. The prevalence of steatosis in HCV-infected patients is 2.5 times higher than in non-infected [6]. Furthermore, liver steatosis occurs in more than 50% of patients with chronic hepatitis C [79]. Interestingly, HCV genotype has an impact on the development of steatosis. In particular, genotype 3 induces more steatosis than others. Furthermore, Huh7.5 cells transfected with genotype 3a contain more neutral lipids in lipid droplets, and larger lipid droplets, than cells transfected with genotype 1b sequences, suggesting that HCV core protein–lipid droplet interaction could play a role in virus-induced steatosis [80]. Importantly, no genetic or functional differences were found between genotype 3a core proteins from patients with and without HCV-induced steatosis, suggesting that other viral proteins and/or host factors are involved in the development of hepatocellular steatosis in patients infected by HCV genotype 3a [80]. An excess of LD is found in many aggressive cancers [81,82,83]. Interestingly, it has been reported that the accumulation of LD in response to acidosis within tumor cells played a major role in triggering the EMT process [84]. Notably, in this last study, the authors showed that prevention of LD formation by pharmacological inhibition of Diacylglycerol O-Acyltransferase 1 (DGAT1) led to a major decrease in the invasion capacity of cancer cells, thus implying that LD would be required in the EMT process involved in this tumor cells invasion [84].

HCV infection of hepatocytes increases the production and secretion of transforming growth factor (TGFβ) by these cells, notably via the core protein of the virus [85]. This massive production of TGFβ participates in the development of fibrosis via the activation of hepatic stellate cells (HSC) [86]. Physiologically, the TGFβ induces the initiation of apoptosis in hepatocytes. However, under certain conditions, such as during HCV infection, TGFβ acts as a strong EMT inducer. This observation should be contrasted with the ambivalence of TGFβ as a tumor suppressor vs. pro-oncogene in the liver [87,88]. HCV modifies the cellular response to TGFβ by several mechanisms. First, by increasing the stiffness of the extracellular matrix. Indeed, it has been shown that an excessively dense matrix induces the resistance of primary murine hepatocytes to apoptosis induced by TGFβ [89,90]. These cells that do not enter into an apoptosis process will instead enter into an EMT process that will participate in their dedifferentiation in a FAK/Src/Akt/Erk/p38 dependent signaling that is over activated on a stiff ECM [89]. Conversely, these same cells cultivated between two layers of flexible collagen I are organized to form a structure similar to bile canaliculi, delimited by MRP2, and their stimulation by TGFβ leads to the triggering of apoptosis. Second, HCV modulates TGFβ signaling more directly, by interacting with effectors of this pathway. Indeed, expression of the HCV core protein in HCC cell lines and in primary murine hepatocytes is able to induce a shift in the cellular response to TGFβ from apoptosis to EMT by interacting with the Mothers against decapentaplegic homolog 3 (SMAD3) protein of this pathway [91]. Finally, HCV core protein stabilizes the zinc finger protein Snai1 and participates in the Snai1/HDACs complex that participates in the induction of EMT [92]. To summarize, HCV infection establishes a deleterious environment for hepatocytes, by disturbing cell–cell junctions, trafficking and ECM. This facilitates the entry of these cells into EMT that could participate in the evolution of the disease as depicted in Figure 3.

## 9. Conclusions

In this review, we have elaborated on the close interactions between HCV and the cell polarization machinery. HCV hijacks the key elements of the polarization processes, including molecules from cell–cell junctions and intracellular trafficking, to enter the cell and replicate. HCV also alters the ECM and modulates the cellular response to TGFβ, thereby participating in hepatocyte transformation and cancer development. Importantly, some of these HCV-induced alterations persist despite clearance of the virus. Thus, work remains to be done to determine the factors involved in the maintenance of these deregulations, in order to propose solutions to minimize the risk of HCC development in cured HCV patients and even in non-cured patients.

## Figures and Tables

**Figure 1 pathogens-11-00337-f001:**
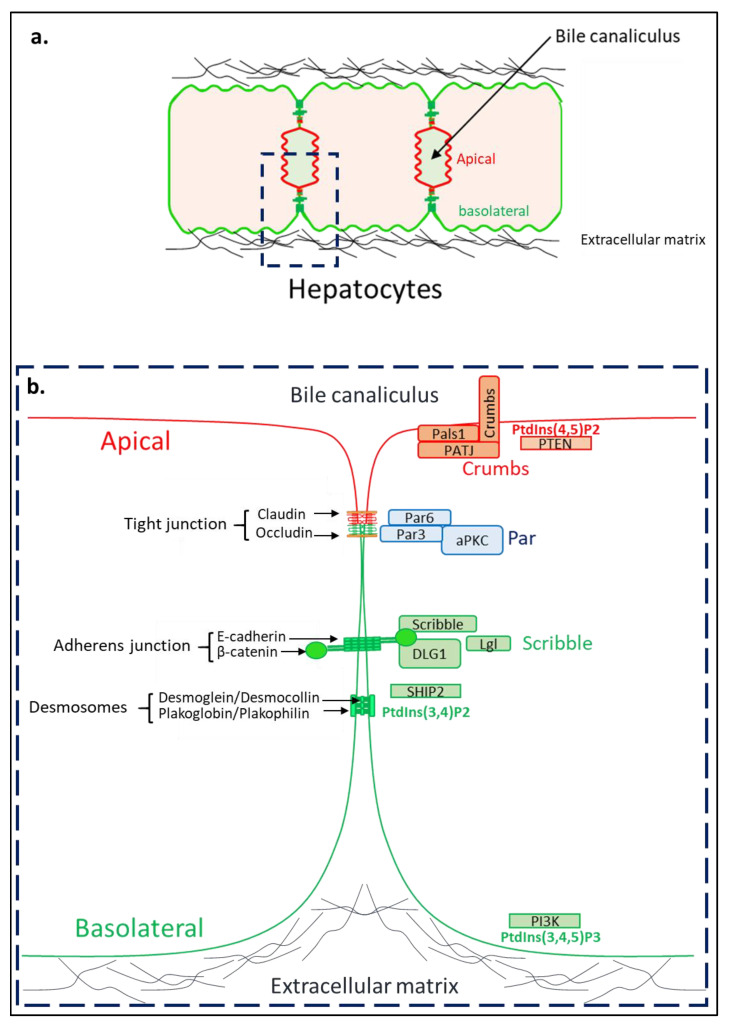
Hepatocyte polarity. (**a**) Within a hepatic lobule, hepatocytes are organized in trabeculae along the hepatic sinusoids and form multiple lumens, the bile canaliculi. Hepatocytes are inserted between two very thin layers of extracellular matrix. (**b**) Enlargement of the framed region in panel a. There are four main types of cell–cell junctions: the tight junctions, adherens junctions, desmosomes and gap junctions. Together, these junctions participate in the sealing and cohesion of the epithelium. The tight junctions, formed by claudins, occludins are associated with the actin cytoskeleton by the adaptor proteins zona-occludens. These junctions ensure the tightness of the epithelium. Adherent junctions made up of cadherins are associated with the actin cytoskeleton by catenins and maintain the architectural integrity of the epithelium, in association with desmosomes. The establishment of cell–cell and cell–ECM junctions initiate cascades of signaling pathways allowing cell polarization, including the establishment of a polarized cell traffic. This signaling is mainly mediated by polarity complexes and enzymes involved in phosphoinotides metabolism. There are three polarity complexes that are essential for the establishment and maintenance of the apico-basal polarity of epithelial cells: Crumbs, Par and Scribble. Among the enzymes metabolizing phosphoinositides involved in apico-basal polarity, we find the proteins PTEN and PI3K and SHIP2.

**Figure 2 pathogens-11-00337-f002:**
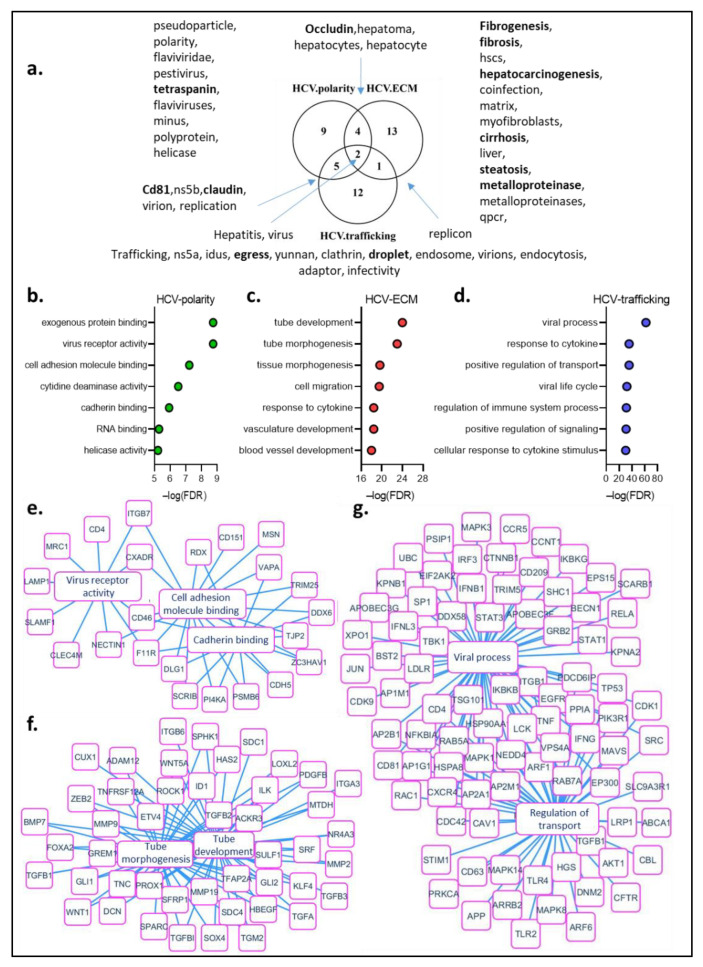
Text-mining analysis for HCV and polarity. (**a**) Venn diagram of keywords found associated with HCV-polarity, HCV-ECM and HCV-trafficking in the Pubmed database. Terms in bold have been described in this review (**b**–**d**) Gene ontology biological processes of the top 100 genes associated by text-mining with both HCV and polarity (**b**), ECM (**c**) and trafficking (**d**). (**e**–**g**) Functional enrichment network of some biological processes found in (**b**–**d**), highlighting a membrane molecular network shared by HCV and cell adhesion (**e**), tube morphogenesis (**f**), and transport during viral process (**g**).

**Figure 3 pathogens-11-00337-f003:**
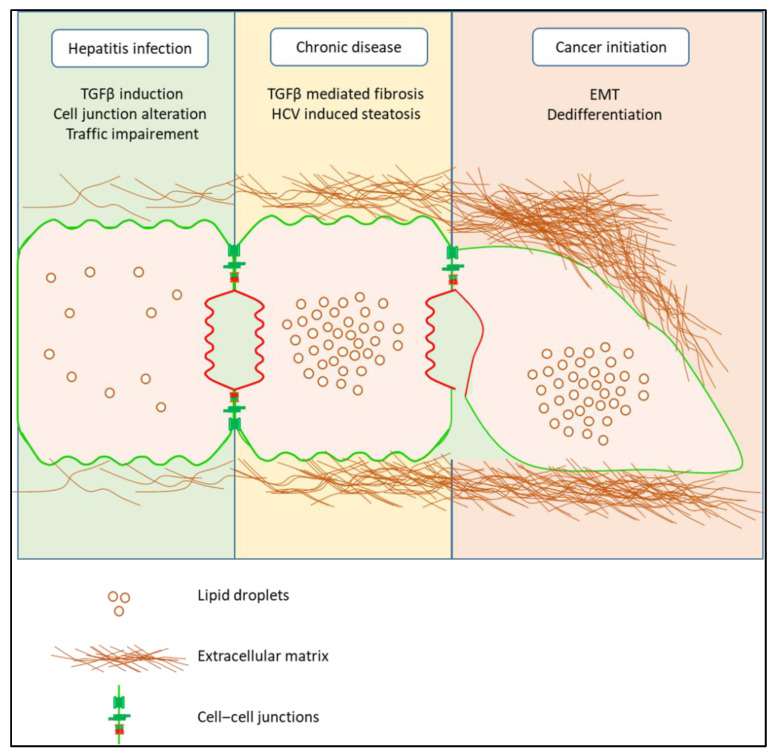
Impact of HCV infection on hepatocyte polarity and differentiation. During primary infection, HCV alters cell junctions, trafficking and stimulates the synthesis and secretion of TGFβ. These alterations contribute to the long-term development of liver diseases such as fibrosis and steatosis. Then, in the very long term, this harmful environment for hepatocytes leads to their transformation, to their dedifferentiation, phenomena that contribute to the development of hepatocellular carcinoma.

## Data Availability

Not applicable.

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
