# Peer review of "Impact of HCV Infection on Hepatocyte Polarity and Plasticity"

_pathogens, 2022, doi:10.3390/pathogens11030337_

Round 1

Reviewer 1 Report

The manuscript by Agnetti et al details the link between HCV infection and its carcinogenic mechanisms. The organization is effective and the text flows well. This reviewer has only one comment about Figure 2. Please insert enough space between figures 2b, 2c and 2d so as to avoid overlap. Figures 2e to 2g have text outside the boxes and sometimes that makes the text difficult to read. Please correct this.   

Author Response

We thank you for the time you spent reading our manuscript and your comments . Please find in attachment our point-by-point response to your comments.

Reviewer 2 Report

In this manuscript, Agnetti et al. review the link between hepatitis C virus (HCV) and cell polarity/plasticity. This review is original as - to the reviewer’s knowledge - the link between HCV, polarity and plasticity has not been reviewed before in this way, likely due to the small number of studies dedicated to this topic. The manuscript could be improved by taking into account the points listed below.

  • It could be added to the introduction that there is no vaccine to prevent HCV infection (contributes to explain why there are 58 million infected individuals and 1.5 million new infections per year).
  • Cite additional studies from the McKeating group (Mee et al. J Virol 2008 and 2009) as well as Randall and colleagues (Baktash et al Cell Host Microbes 2018).
  • Information (a chapter?) on HCV cell model systems (Huh7 vs HepG2, 2D vs 3D) could be included to discuss findings obtained in different models that are or are not polarized.
  • Line 151 and following: additional details about the role of junctional vs non junctional claudin-1 should be added in order to clarify the role of this protein in HCV entry. In liver-derived cells, the non-junctional localized protein appears to be targeted by an anti-claudin-1 antibody inhibiting HCV entry (Mailly et al. Nature Biotechnology).
  • What is the link between chapters 4 and 5? An introductory sentence should be added to chapter 5 in order to improve the flow.
  • There should be a conclusion for chapter 6. What are the consequences for HCV infection and/or HCV-induced pathogenesis?
  • Chapter 7 does not seem very informative if text-mining study only refers to the co-occurrence of the terms “HCV” and “polarity” in a document. This part could be removed.
  • The layout of illustrations is very basic and could be improved. Figure 1: it should be clarified in the figure that the lower part is a zoom on the framed portion of the top part, e.g. by adding dotted lines from the frame to the edges of the lower part as often done for microscopy images.

While the text is overall well understandable, English Grammar and spelling should be polished and the manuscript carefully proofread. Some parts that need attention are listed below:

  • Abstract, line 11: “nevertheless” rather than “although”?
  • Line 42: it should read interferon alpha (not gamma)
  • Line 45: the study from Gal Tanamy and colleagues should also be cited (in addition to reference 8 and 9) and a few details given regarding the underlying mechanisms of these persisting modifications induced by HCV and contributing to HCC risk.
  • Line 61: “positive strand RNA virus” instead of “RNA-positive oncogenic virus”
  • Line 158: “their” instead of “there”
  • Line 221: “of note” rather than “to note”
  • Line 225: the reference is not properly formatted
  • Line 250: “occludin” not “occluding”
  • Line 279: “cytoplasm” instead of “cytoplasmic”
  • Line 285: rephrase - “in more than 50%”

Author Response

(The authors gave the same response as above.)

Reviewer 3 Report

This review article deals with a particular topic, which is not well covered by recent reviews in the field.

1.The topic of changes in hepatocyte polarity due to HCV may be considered irrelevant, if HCV is curable. Which of the changes persist after SVR? What are the underlying mechanisms (e.g. epigenetics)? – The authors are not very clear on this aspect. Possibly, a table summarizing the most relevant changes, their durability, molecular mechanisms and the practical consequences (e.g. HCC risk) would be helpful.

2.To which extent are the mechanisms influenced by HCV genotypes? For instance, steatosis is much more common in HCV GT 3 patients. The mechanisms regarding lipid droplet formation are likely influenced by HCV genotypes, the authors should explain how and why.

3.Many of the described changes – e.g. bile acid metabolism, TGFb secretion – will have consequences of relevance for liver health. For instance, TGFb will impact immune cells (e.g. Kupffer cells) and stellate cells. Bile acids may reach the gut and modulate the microbiome. The authors are not elaborating on cross-talk of HCV-infected hepatocytes with other cells in the liver, the immune system and extrahepatic organs.

Author Response

(The authors gave the same response as above.)

Round 2

Reviewer 2 Report

The authors have taken into account the reviewer's comments  in order to improve their manuscript.

Reviewer 3 Report

My comments have been addressed.